# Investigations on Temperatures of the Flat Insert Mold Cavity Using VCRHCS with CFD Simulation

**DOI:** 10.3390/polym14153181

**Published:** 2022-08-04

**Authors:** Rong-Tsu Wang, Jung-Chang Wang, Sih-Li Chen

**Affiliations:** 1Department of Tourism and Leisure Management, Yu Da University of Science and Technology, Miaoli County 36143, Taiwan; 2Department of Marine Engineering (DME), National Taiwan Ocean University (NTOU), Keelung 202301, Taiwan; 3Department of Mechanical Engineering (ME), National Taiwan University (NTU), Taipei 10617, Taiwan

**Keywords:** plastic, injection molding, simulation, vapor chamber, heating and cooling system

## Abstract

This paper adopted transient CFD (Computational Fluid Dynamics) simulation analysis with an experimental method for designing and surveying the quick and uniform rise in the temperature of the plastics into the insert mold cavity. Plastic injection molding utilizing VCRHCS (Vapor Chamber for Rapid Heating and Cooling System) favorably decreased the defects of crystalline plastic goods’ welding lines, enhancing the tensile intensity and lowering the weakness of welding lines of a plastic matter. The vapor chamber (VC) possessed a rapid uniform temperature identity, which was embedded between the heating unit and the mold cavity. The results show that the tensile strength of the plastic specimen increased above 8%, and the depths of the welding line (V-gap) decreased by 24 times (from 12 μm to 0.5 μm). The VCRHCS plastic injection molding procedure can constructively diminish the development time for novel related products, as described in this paper.

## 1. Introduction

Globally, the used capacity of plastic first became greater than steel after the 1980s [1,2]. Injection molding was the most frequently applied manufacturing procedure and one of the most important techniques for global fabricating plastic parts [3,4]. However, it was a technologically complicated procedure and very hard to consider all the transformations occurring during the procedure. These complications include varying temperatures, filling velocity and pressure, heating and cooling times, and packing and injection times [5,6,7]. A wide variety of goods can be made utilizing injection molding [8]. Plastic materials have been widely employed in machinery, the automobile industry, electrical, optical, medical, and food. They are also used in other industries, including mobile phones, computers, high-precision optical fiber connectors, CD films, and light guide plates. This is due to the improvement in the quality of plastic materials and the development of types [9,10,11]. These products have good electrical and mechanical properties and are lightweight and easy to process with corrosion resistance and no-conductivity [12]. The injection molding process had the advantages of fast production and automation. It was one of the most widely used processes in producing plastic products [13]. The plastic was melted through the screw of the injection molding machine and injected into the mold. The plastic cools and solidifies to make a product. In the injection molding process, the plastic flows in a laminar flow, and the melt front fills the mold cavity in the form of a fountain flow [14,15]. The fountain flow allows the plastic to gradually solidify from the contact side of the mold wall to the center layer of the plastic [16]. The increased solidification rate of the consolidating layer accelerated as the mold temperature was too low. The passage area for the melt was reduced, increasing the melt flow resistance [17]. When the plastic was difficult to flow during the filling stage, it caused appearance defects (surface finish, roughness, and sink marks) and had the problems of warping and deformation due to uneven residual thermal stress and volume shrinkage [18,19]. Therefore, suitable operating parameters were important for improving the effectiveness of the process and consistent product quality. The operating parameters include injection speed, mechanical properties, holding pressure, melt temperature, mold temperature, injection time, holding time, and cooling time. Although the injection molding process had certain advantages in production, the appearance defects of plastic products involving wind wrapping, burrs, appearance scratches, short shot, welding line, warpage caused by residual stress, and poor precision were still problems. The manufacturers must invest a lot of time and money to solve the process of product development and production under the constraints of product modeling, mold structure, and machine adjustment [20,21].

Most researchers have investigated injection molding process parameters and established some correlations during production to assure quality and minimize defects [22]. Among these problems, welding lines were one of the most common cosmetic defects. Although this defect can be overcome by secondary processing, such as painting and electroplating, manufacturers want to avoid painting or electroplating as much as possible during product development. Manufacturers especially avoid these solutions where a large-scale application of plastic transparent appearance parts (such as Apple’s computers), plastic recycling restrictions caused by environmental protection issues, and product cost control are involved. Moreover, this demand also made the appearance defect of the welding line become the most troublesome problem for product developers when designing the appearance. The main reason for the welding line defect was that the air at the plastic junction could not be discharged from the mold cavity [23]. It was often impossible to increase the exhaust in the mold when making transparent plastic parts, resulting in the more obvious traces of the welding line. From the experience of previous molding personnel, increasing the mold temperature or the plastic molding temperature assisted in eliminating the welding line under the limitation that there is no additional exhaust mechanism. The major method used was to increase the mold temperature (increasing the plastic molding temperature can easily cause the plastic to crack) in the above two methods. Sánchez et al. [24] introduced the mold temperature curve to understand how RHCM (rapid heat cycle molding) works and compared the results with conventional molding. They studied an electrical power system for a testing part injected with two amorphous polymers (PC and ABS) under conventional and RHCM systems using a pressure–temperature transducer and IR (infrared camera) to record both parameters in the injection molding cavity. Measured values and tendencies were compared with simulation tools with good correspondence. They considered that surface temperature curves simulated and measured for both processes and thermal inertia observed in the heat and cool processes should be carefully premeditated at the beginning of the plastic part project. Li et al. [25] presented a novel method for predicting the warpage of crystalline parts molded using the RHCM process. The results showed that the microstructure and temperature were symmetrical along the thickness direction in CIM and asymmetrical in RHCM. The predicted warpage was influenced by crystallinity. The warpage predicted with crystallinity was larger than the warpage without crystallinity, especially in the parts molded by RHCM. The proposed method was accurate and effective. It was a potential candidate technology for quantitatively predicting the warpage of plate parts and optimizing the molding process for manufacturing. Poszwa et al. [26] investigated the influence of RHCM on the basic parameters of filling the cavity using numerical analysis. Obtained results showed strong nonlinear relations between maximum flow length, part thickness, and volumetric flow rate. Whereas the relation between flow length, injection pressure, and cavity surface temperature was linear.

Liparoti et al. [27] carried out the thermal transient simulation with very thin multilayer heating devices layered on the cavity surface for an injection molding process. A conductive layer made these thin heating devices between two insulating layers with thicknesses. They were layered on the cavity surface, allowing temperature evolution between injection and cooling channel’s temperatures to be as fast as possible. The results of the simulations (temperature and pressure distribution evolutions) were compared with experimental results and were successfully justified. An increase in the mold temperature and a high mold temperature reduced the viscosity of the plastic entering the mold cavity. A better fluidity reduced the forming pressure and the clamping force. However, the time required for the plastic to achieve ejection temperature from the melting temperature was inevitably increased as the mold temperature and the forming cycle time increased. Bianchi et al. [28] compared a novel approach based on cooling rate optimization with RHCM techniques in ceramic injection molding. Results showed that the new method emerged with the benefits of higher dimensional control and reduction of differential shrinkage compared to the other analyzed approaches. This resulted in an increased capability to use injection molding to manufacture ceramic components characterized by non-uniform wall thickness. Cho et al. [29] designed a 3D printed insert core to improve weld line defects on an automotive crash pad with a high standard of appearance by local heating and cooling. Results showed that it is more efficient than other RHCM techniques because the 3D printed insert core does not need additional equipment or high energy consumption. Kitayama et al. [30] adopted a radial basis function network to determine the optimal multi-objective process parameters in rapid heat cycle molding (RHCM) for improving the weld line, clamping force, and cycle time. The numerical and experimental results clarified that the weld line reduction could be achieved by the proposed RHCM. Fu et al. [31] proposed a novel ejection criteria model based on the partial solidification for early ejection in plastic part molding. They investigated the general time determination model targeting to lessen the cycle time. They compared the traditional ejection criterion of the fully solidified by in-mold cooling. The results revealed that the cycle time could be decreased significantly. Nevertheless, increasing the mold temperature will prolong the molding time required for the plastic part (the plastic part must be cooled to the ejection temperature in the mold before the mold can be opened), resulting in unnecessary costs. Poszwa et al. [32] investigated the injection molded flexible hinge parts manufactured with selective induction heating to improve their properties. The linear relation between the number of cycles the hinges can withstand, mold temperature, and injection time was identified. The mold temperature was a more significant factor. Monti et al. [33] studied the influences of distinct ethylene copolymers in raising the collision strength of a fiber-reinforced composites based on a recycled rPET from post-consumer bottles. This research indicated that a post-consumer PET from city waste could be utilized as a high-performance pattern. Injection-molded composites are appropriate for application in the automotive sector with no agreement regarding mechanical demands or thermal steadiness. Verma et al. [34] first designed different lattice structures in a CAD environment. The strength of the subsequent lattice-resin interlock was predicted using the finite element analysis (FEA) method. Individual stress status of lattice and resin during the tensile test were investigated to determine the failure strength of lattice-resin interlock. FEA predicted failure strength was correlated with the experimental result with reasonable deviations. Xu et al. [35] presented a detailed study on the compression-molding process of micro-grooves on the carbon fiber-reinforced plastic (CFRP) surface. They successfully obtained the appropriate parameters. The experimental results showed that the micro-groove array structures on the CFRP surface could effectively improve the composite parts’ tensile strength of the connection interface.

These general defects were often discovered on the external appearances of welding lines. They are conformation shortcomings and also lower the mechanical intensity of the assemblies. A welding line may be taking shape for the plastics by moving around the inserts as the metal embeds were put in the mold for injection molding. Conventionally, the inserts were placed in the mold at the surrounding temperature, yet the inserts’ temperature was inferior to that of the mold, as the filling was executed due to mold sealing. Therefore, this paper attempted to design and present a set of local rapid heating mechanisms (e.g., RHCM and VCRHCS) at the meeting point of the welding line of the molded product. The design considered the procedure parameters and effect on the quality of goods and used the local increase of the mold temperature to lower the viscosity of the plastic. This is because the plastic can heighten the fluidity of the plastic when it meets, and the welding line traces can be eliminated. Throughout this study, the procedure parameters, reaction, the material employed, and the technology applied to the best research in the field of injection molding were highlighted. 

## 2. Methodology and Materials

The experimental framework of this study was divided into four categories: heating mechanism design, temperature measurement and heat transfer simulation, welding line strength experiment, and welding line appearance experiment. First, a heating mechanism was designed by lever contact and separation with the VC (vapor chamber) for the temperature measurement. The results were compared and verified with the CFD simulation analysis. Tensile test pieces and holes in flat mold cores were used to make test specimens employing an injection molding machine. They were subjected to tensile tests, appearance observation, and SEM (scanning electron microscope) microstructure observation. Figure 1 displays the complete experimental architecture diagram. Figure 2 displays a vapor chamber (VC) [36,37] with superior thermal performance and a two-phase (vapor–liquid) heat transfer device [38]. High-performance servers, high-end VGA, and high-power LEDs were employed [39,40]. The general operating fundamental of a vapor chamber is explained. The inside of the VC has a vacuum, and then the working fluid will be quickly converted into vapor by boiling or evaporation after the wall face of the VC cavity assimilates the heat from the heat source [41,42,43]. Finally, the vapors will be condensed into liquid because of the cooling resulting from the heat convection on the external barrier of the VC cavity and restream to the area of the heat origin along the wick structure [44,45]. Therefore, VC propagates huge heat capacity speedily and is achievable to be adopted in insert molding manufacture. The metal insert was put into the mold initially and fabricated into inserted plastic goods via injection molding [46,47]. The dimensions of the VC employed in this study were 80 × 50 × 4 mm^3^ with a porosity of 0.46 and the lowest thermal resistance of 0.15 °C/W. The goods can be created in a single molding procedure, reducing the processing time and decreasing the possible manual mistakes emerging from some processes.

The installation of VCRHCS adopted a vapor chamber to promote the heating and cooling apparatus for the injection molding procedure, as shown in Figure 3. A current heating structure of the VC required that the mold have a movable slider at the interior area installed with two heating rods. The VC was fixed at the union of the front side of the slider and the mold barrier. The mold temperature was increased upon the glass transition temperature of the plastic before the filling phase. Then, the cooling of the mold was initiated at the packing phase. The operation sequence of VCRHCS is that the heated slider will move ahead and touch the VC. As the embeds are placed in the mold, the mold is locked, and the inserts are warmed instantly with the VC. Subsequently, the heating cycle is operated by a pedal mechanism composed of a hydraulic cylinder. It presses the VC to touch the mold at the filling phase. A heating and cooling apparatus with VC was exploited in the present study. A VC was mounted between the heating block and the mold cavity. The dimensions of the heat source of VCRHCS were 50 × 50 × 80 mm^3^. It can be combined with the mold in any situation, regardless of the mold’s magnitude. The mold steel block of P20 was heated by heating rods constituted with two electrical heating tubes. These tubes touched VC when the pedal was functioning, and the T-type thermocouples were inserted for recording the temperatures of the facilities. The heat source was disconnected from the VC when the filling was accomplished.

In addition, Figure 3 revealed five thermocouples arranged on the outside of the cavity for gauging the temperatures of the vapor chamber. The central temperature of the cavity and VC was metered at Point O. The temperatures of the opponent to the locations of the heat embed apparatus and VC on the outside of the cavity were metered by Points A and B and Points C and D, respectively. Point C was the farthest from point O. The photograph of the whole heating and cooling apparatus is shown in Figure 4. The weight of this mold of VCRHCS is over several hundred kilograms. The magnitude and capacity of the heating and cooling system can be altered to adapt to the mold. In this experimentation, the plastic was ABS (Chi-Mei PA-758) with a glass transition temperature of 109 °C. The mold base and mold core materials were JIS S50C and ASSAB 718, respectively. There were two injection molds judged the flat goods utilizing VC, and the testing material was ABS. The VCRHCS can enhance the tensile intensity and decrease the flaw of welding lines of plastic goods because of fast and consistent heating and cooling recurrence with VC.

Before the VCRHCS experimental verification, the VC thermal performance of CFD (computational fluid dynamics) simulation analysis, in which fluid mechanics, discrete mathematics, numerical method, and computer technology were integrated by applying computer-aided design (CAD) and analytical software tools (Ansys Icepak 2022R1, Canonsburg, PA, USA). The relationships of the rapid uniform temperature characteristics at unsteady state conditions were coordinated to predict and control the heat source temperatures during the heating stage. Figure 5 demonstrates the simulation analytical models of the present device. Figure 5a shows a no vapor chamber system meaning that the heat source directly touched the P20 mold. Figure 5b is a VC system. Consequently, the VC was placed between the heat source and P20 mold. The heat flow field of VC was simulated transiently through numerical analysis, and a contrast was depicted between the experimental and simulation results. The numerical analysis procedure in this simulation can be separated into pre-processing, numerical solving, and post-processing. They had the same simulation conditions, power, and size of the heat source. All conditions of CFD models were the same except when using VC or not. CFD can reduce many costs of manufacturing and rapid design. It can verify the thermal performance of VC within a short period. The entire simulation analysis range was 250 × 150 × 150 mm^3^. The dimension and input power of the thermal source were 20 × 25 mm^2^ and 130 °C, respectively. The size of VC was 80 × 50 × 5 mm^3^ with a thermal conductivity of 700 W/mk, density of 8900 kg/m^3^, and specific heat of 2.283 kJ/kg°C. The size of P20 plastic mold steel (ASSAB 718) was 160 × 80 × 50 mm^3^, which had a better thermal conductivity of 29.5 W/mk. The boundary conditions and physical heat properties involved the environmental temperature of 23 °C. The whole simulation analysis model was a transient situation from 0 to 60 s. There were about 1500 thousand grid elements, a 0.01 s time step, and 200 iterations in the present study. Each simulation took about 4 h.

## 3. Results and Discussions

Figure 6 reveals top to bottom temperature transfer changes for the simulation results of the heat transfer between the process with a vapor chamber and the process without a VC at 1, 20, 40, and 60 s, respectively. At the beginning of heating at 1 s, the rapid uniform temperature characteristic of VC was demonstrated. The ranges of heat transfer using a VC were larger than that of the case without the VC, and the temperature distributions were also relatively uniform from the temperature lines and colors distribution. The simulation results of the two points of A and B were all 82 °C at 60 s with VC. The temperatures of the two points of A and B were all 51 °C at 60 s without VC. Figure 7 displayed the temperature profiles of 0 to 60 s with VC and without VC. The temperatures of Point O and the farthest side from Point O to point C were 74 °C and 35 °C under no VC and 60 s. Moreover, the temperature variances of the other points of A, B, and D were great morals around 10 °C each other at 60 s and without VC. However, the temperatures of Point O and Point C were separately 84 °C and 68 °C at 60 s with VC. Moreover, the temperatures of the other points of A, B, and D were nearly 80 °C at 60 s with VC (simulation results of 82 °C). The temperature profile of Point O without VC was raised faster than that with VC between 0 s to 30 s due to the heater’s immediate touch but decelerated after 30 s. Point O reveals the central temperature of the cavity and VC. In summary, the temperature of Point O without VC was higher than with VC before 30 s. However, the slope of the temperature of Point O with VC was high than that of Point O without VC. The mean temperatures of these five points with VC and without VC were 78 °C and 56 °C based on the experimental results. In the simulation results they were 81 °C and 60 °C, separately. The high-speed uniform temperature influence of the employed VC heating system (VCRHCS) was beneficial, although the heater touched Point O on the outside of cavity with no VC apparatus.

Two distinct mold designs, including one gate mold and two opposite gates mold in the present experiments, applied VCRHCS in tensile testing pieces, as shown in Figure 8. The tensile test samples were one and two gates, as shown in Figure 8a. The parts of the injection molding experimentations were tensile inspecting pieces. They assessed the successfulness of the heating and cooling system accompanied by a vapor chamber with no VC. The welding line was in the central position, and the tensile sectional area was smooth, as shown in Figure 8b,c. However, the cut sectional area was sharp for one gate tensile testing piece because there was no welding line in the central place. as shown in Figure 8d. Figure 9 exhibited testing pieces SEM images of welding line and the relationships of the temperatures with the heating time with/without VC. If there were no influence of VC, these welding lines are clearly displayed under the SEM photographs in Figure 9a. One gate piece had no welding line in the central position, as shown in Figure 9d. Raise the heating temperature, less obvious the welding line for using the VCRHCS. Figure 9b had a slight welding line at a heating temperature of 75 °C. Figure 9c had almost no welding line at a heating temperature of 110 °C. The dimensions of these tensile testing pieces were 215.9 × 12.7 × 25.4 mm^3^ with a thickness of 3.175 mm. Table 1 shows the tensile intensity of the testing pieces in the distinct status of one gate and two opposite gates with and without VC and a 0.5% error. The tensile intensity theorizes the highest value for the one gate system due to no welding line in the central place from Figure 9d. There were no defects of welding lines in the tensile testing pieces from the no VC one gate system. The two gates with no VC heating apparatus showed obvious welding lines. Its tensile intensity dropped by 11.1%, more than the piece with one gate from Table 1 and Figure 9. The two gates with VC heating apparatus showed a tiny welding line, and its tensile intensity of testing piece was better, at 6.8%, than the piece with two gates and no VC apparatus. By raising the warming temperature from 75 °C to 110 °C on the piece with two gates and a VC apparatus, the tensile intensity increases by 3.2%. This is because the VC can dissipate heat rapidly and uniformly transfers heat to the molding steel of P20. The regional increase of the mold temperature lowers the viscosity of the fluidic plastic, and the plastic could ameliorate the fluidity and the welding line marks could be extirpated.

Another testing multi-hole specimen was applied VCRHCS in eight holes pieces to assess the successfulness of the system with vapor chamber in the distinct status of cavity and core temperatures. These multi-hole pieces had eight holes with four 10 mm and 5 mm diameter holes, separately. The dimensions of the eight holes pieces were 110 × 53 × 3.175 mm^3^, as shown in Figure 10. Three temperature associations (60 °C, 80 °C, and 130 °C) were evaluated in the present experiments. Case 1 was the requirement of a cavity temperature of 60 °C and core temperature of 60 °C. Case 2 was the requirement of a cavity temperature of 60 °C and core temperature of 130 °C. Case 3 was the requirement of a cavity temperature of 80 °C and core temperature of 130 °C. Figure 10 illustrates the criterion of the eight holes testing piece and the SEM images of the V-notch. There were many welding lines on the appearance of the diaphanous pieces, especially in Case 1 of Figure 10a. The V-gap was deeper than the welding line and was more evident for transparent products. The depths of the V-gap seen in every case were 12 μm, 2 μm, and 0.5 μm. The piece from Case 1 exhibited a V-notch 24 times deeper than the piece from Case 3. The effects of cavity and core temperatures were also significant for the welding line when adopting the VCRHCS in the present work. Finally, employing VCRHCS demonstrated that the temperature distinctions of cavity adopting VC were smaller than that without VC. Raising preheating temperature can increase the tensile intensity for two opposite gates due to elongating sufficient fluid stream and reducing the fluid’s viscosity. The novel VCRHCS can effectively lessen the deepness of the V-gap by 24 times, resulting in the rapid uniform temperature characteristic of VC in this paper.

## 4. Conclusions

This study employing a VCRHCS (Vapor Chamber for Rapid Heating and Cooling System) certified that the inserts’ temperature values and distributions significantly impact the finishing and gathering strengths of goods among existing insert molding procedures. The rapid uniform temperature distributions of the inserts mold were consistent and verified between the experimental and CFD simulation results. Moreover, a VCRHCS with rapid uniform heating and cooling cycle was able to successfully lower the defects of the welding lines of transparent plastic goods. It was even able to enhance the tensile strength of injection molding products. The hydraulic cylinder controlled a pedal mechanism to push the VC down to touch the mold during the filling phase while heating. The heating temperature of the mold should not only reach the glass transition temperature of the ABS plastic material (Chi-Mei PA-758, 109 °C) but should have a better effect on eliminating the bonding line at a higher temperature (over 109 °C).

The simulation results were consistent with experimental results for the thermal performance of VC. The experimental results indicated that the depths of the welding line of the testing pieces with eight holes in the plastic plate were reduced from 12 μm to 0.5 μm. The tensile strengths of the other plastic testing pieces with two opposite gates were enhanced to 6.8% and 10% compared to the conventional goods. The tensile intensity of the testing piece with two gates and VC heating was only smaller by 1% than the piece with one gate. Installing the VCRHCS on both the male and female mold sides was effective in lowering the defects of the welding lines and accomplishing giant material strength than on one side only. In the process of experiments and transient simulations in this study, it can be concluded that the use of VC for injection molding mold heating and cooling is better than the unused cases in this research.

## Figures and Tables

**Figure 1 polymers-14-03181-f001:**
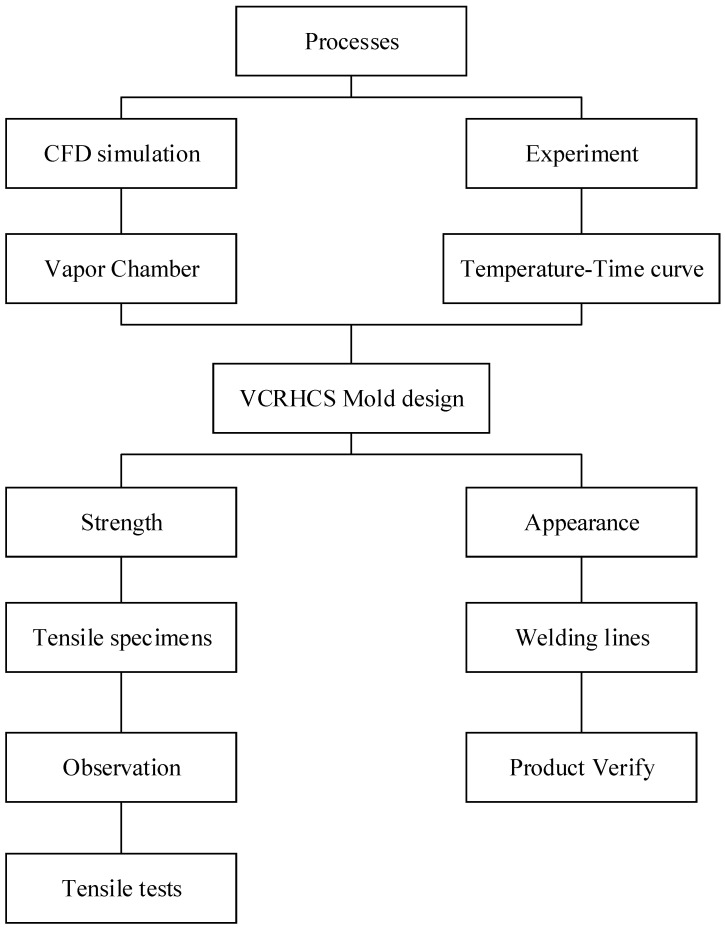
Experimental architecture diagram.

**Figure 2 polymers-14-03181-f002:**
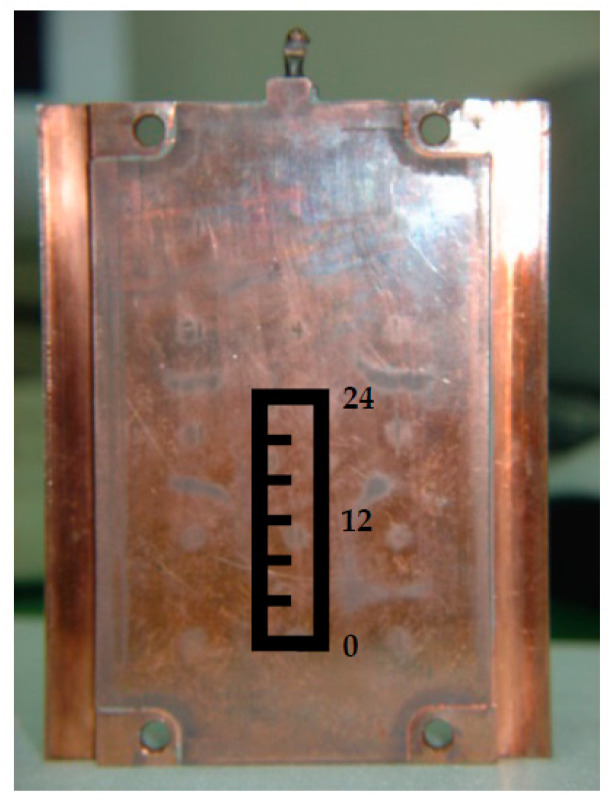
Photo of vapor chamber (VC).

**Figure 3 polymers-14-03181-f003:**
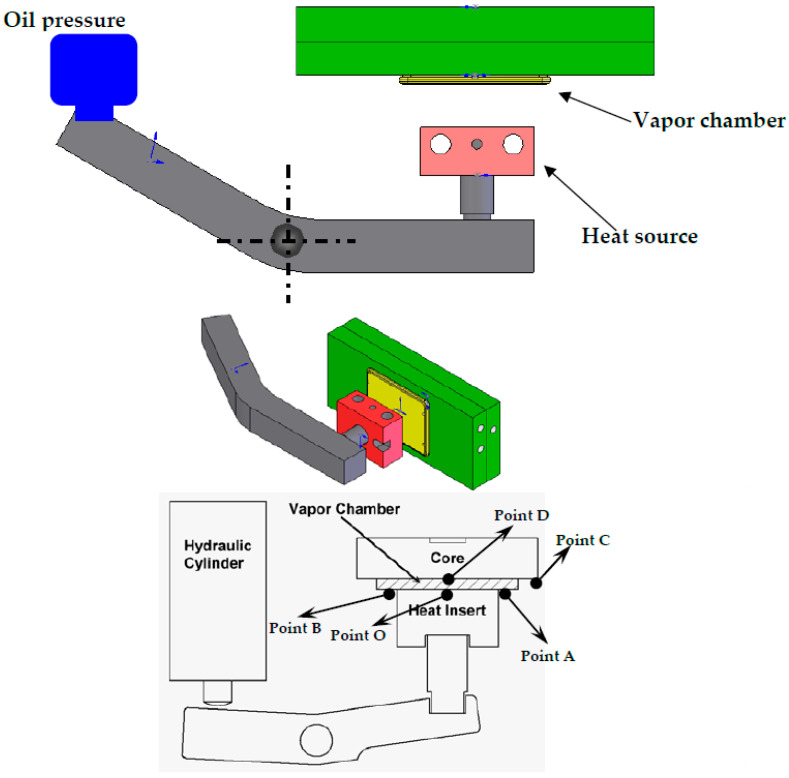
Mechanisms of heating and cooling cycle system with Vapor chamber (VCRHCS).

**Figure 4 polymers-14-03181-f004:**
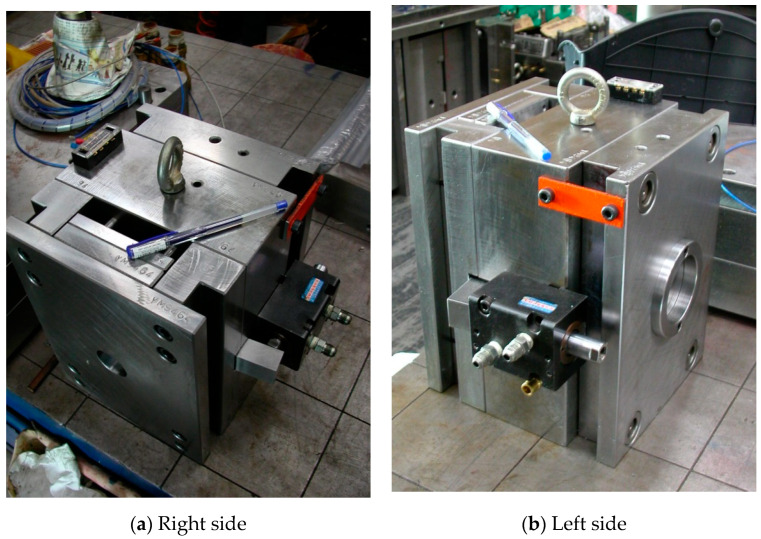
Actual photograph of this mold of VCRHCS.

**Figure 5 polymers-14-03181-f005:**
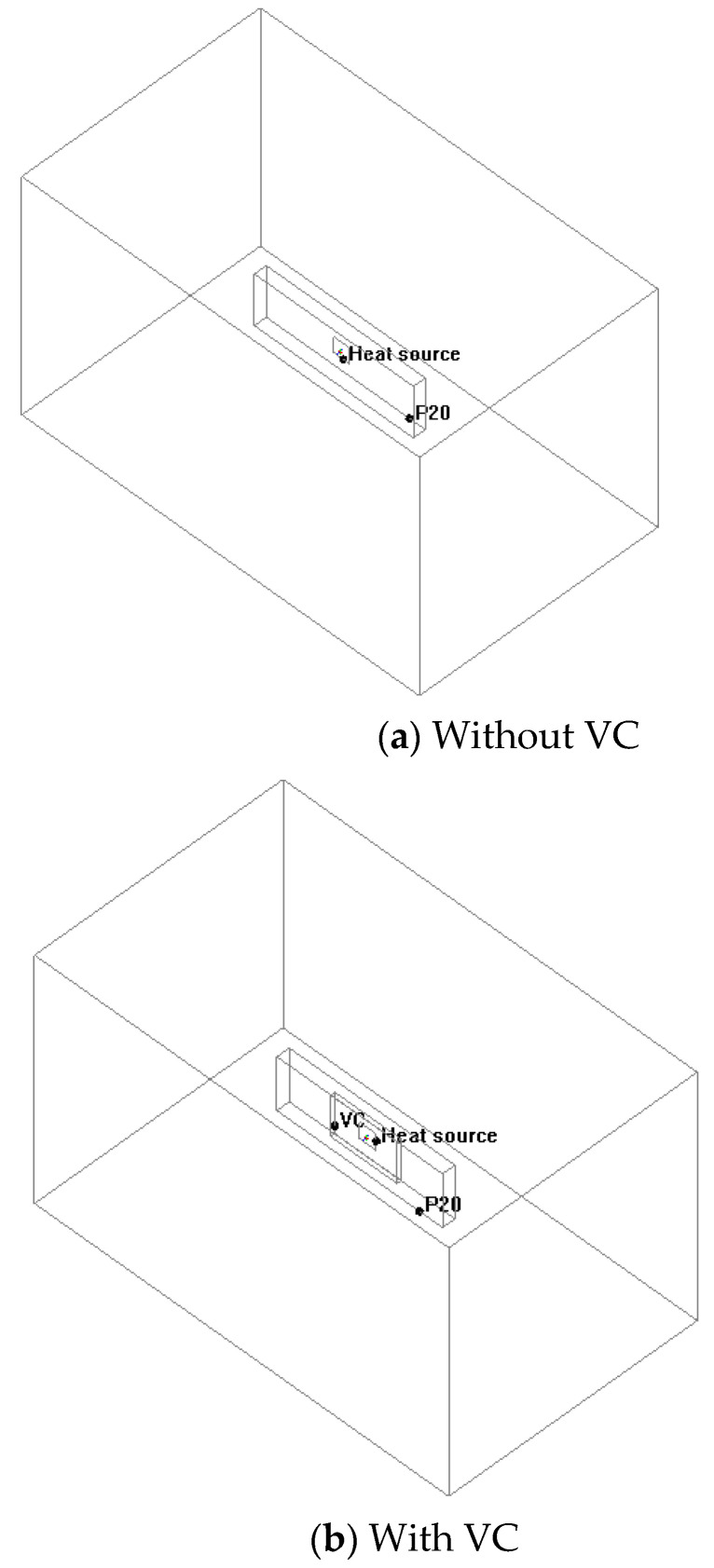
CFD simulation models.

**Figure 6 polymers-14-03181-f006:**
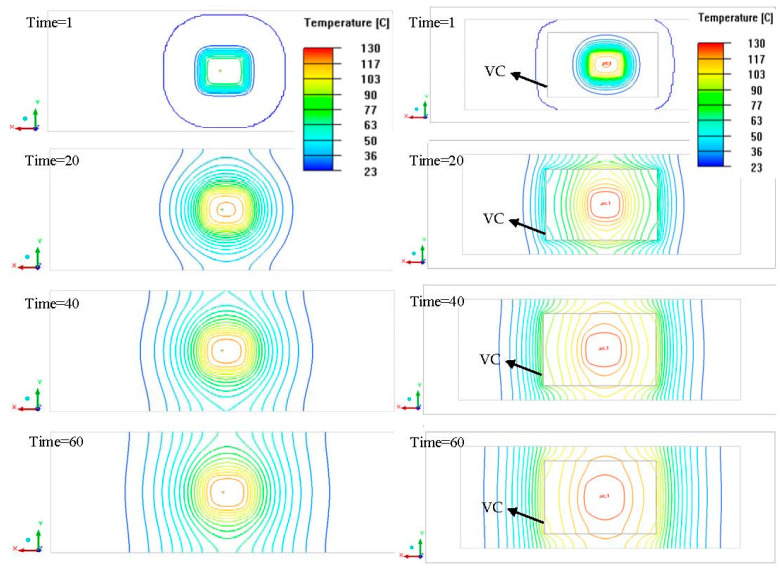
CFD simulation results.

**Figure 7 polymers-14-03181-f007:**
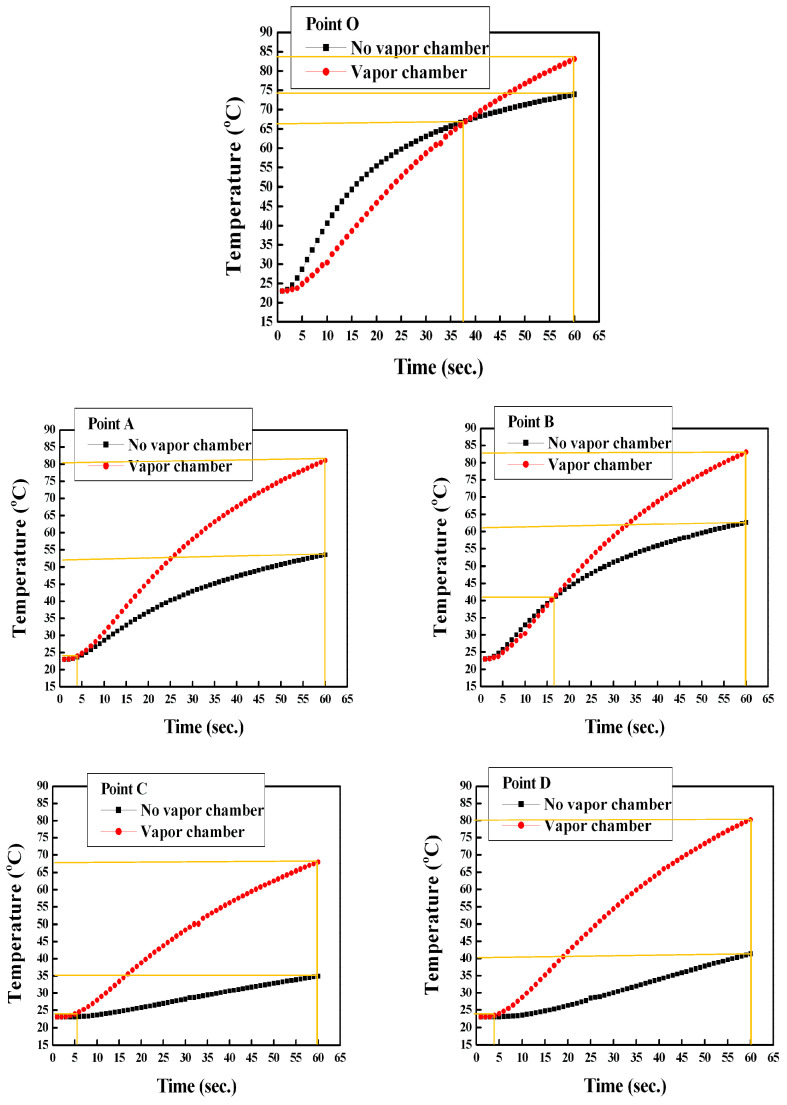
Relations of the temperatures with the heating time with/without vapor chamber.

**Figure 8 polymers-14-03181-f008:**
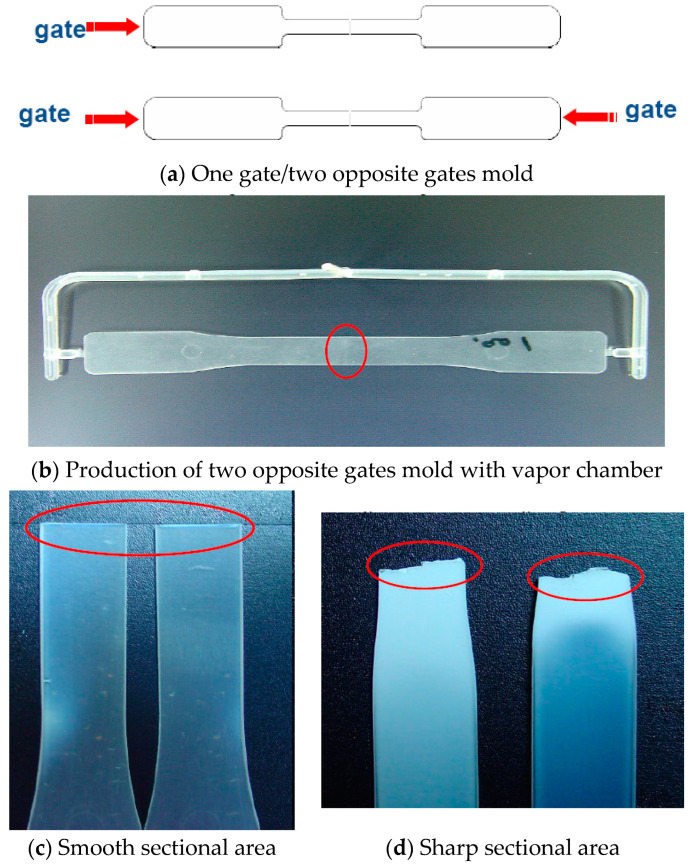
One gate mold with no VCRHCS and two gates mold with VCRHCS.

**Figure 9 polymers-14-03181-f009:**
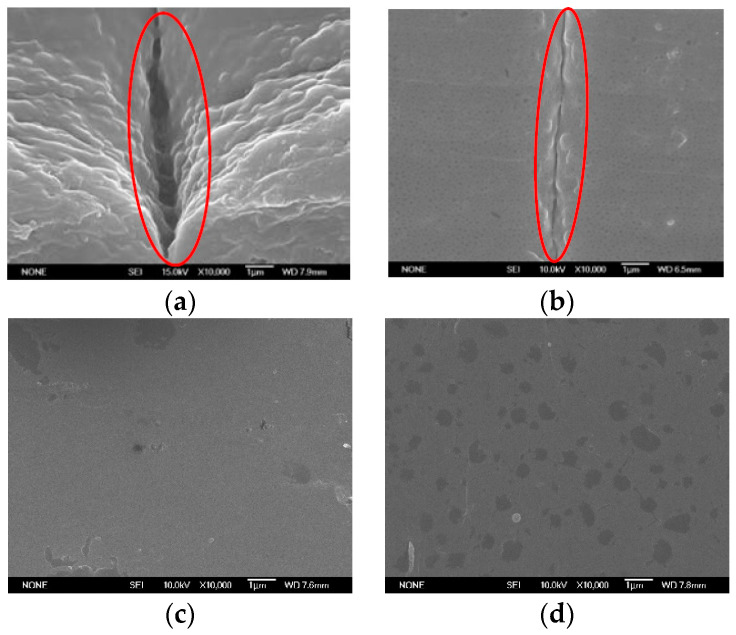
SEM photographs of one gate and two opposite gates. (**a**) Two opposite gates under Point O at 75 °C and with no VC apparatus. (**b**) Two opposite gates under Point O at 75 °C and with VC apparatus. (**c**) Two opposite gates under Point O at 110 °C and with VC apparatus. (**d**) One gate under Point O at 75 °C and with no VC apparatus.

**Figure 10 polymers-14-03181-f010:**
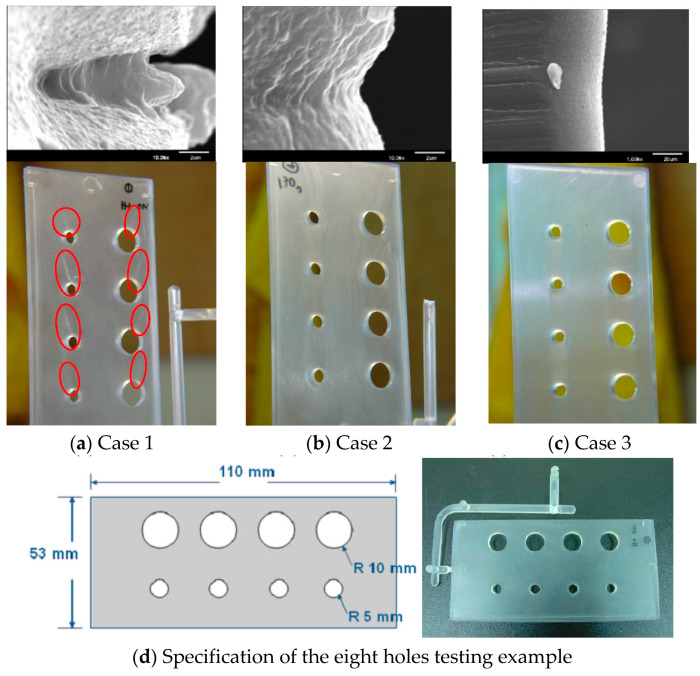
SEM photographs of the eight holes plate.

**Table 1 polymers-14-03181-t001:** The achievements of the tensile examination.

Requirement	Max. Stretch Tension (Kgf/cm^2^)	Intensity (%)
Two gates, no VC Cavity temp. = 75 °C	165	88.9
Two gates, with VC Cavity temp. = 75 °C	178	95.7
Two gates, with VC Cavity temp. = 110 °C	184	98.9
One gate, no VC Cavity temp. = 75 °C	186	100.0

## Data Availability

All data are offered by the authors on reasonable request and the VCRHCS are available from the authors.

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
