# Peer review of "Investigations on Temperatures of the Flat Insert Mold Cavity Using VCRHCS with CFD Simulation"

_polymers, 2022, doi:10.3390/polym14153181_

Round 1
Reviewer 1 Report
The use of VCRHCS could be interested in injected moulding defects control parts. The authors do not explain the behaviour of the study in complex geometry parts. What happens in the case of curved pieces? Does the same behaviour exist?
Author Response
Response to Reviewer 1 Comments
The author is grateful to the reviewers whose comments have helped me improve the manuscript. Thanks Reviewers for their time and effort for the present manuscript. I appreciate his/her comments that my work presents “The use of VCRHCS could be interested in injected moulding defects control parts.”. I have reconstructed and reorganized the whole manuscript carefully. And in order to improve the quality of the paper, the whole manuscript has been checked carefully to avoid any grammar or syntax error. Meanwhile, some statements have been rephrased or refined to improve the language usage in the manuscript. I have accounted and discussed the phenomenon more details in the revised version. I have added more comments to explain the physical effects of the plots. The review comments and descriptions of revisions are listed as following. I have included a separate copy of the revised paper in which I highlighted all the modifications made according to all the reviewers’ comments in RED colour. I believe that the present paper is now acceptable for publication.
Point 1: The authors do not explain the behaviour of the study in complex geometry parts. What happens in the case of curved pieces? Does the same behaviour exist?
Response 1: Thanks for the reviewer’s comment. The paper just discussed the plane surfaces of pieces. If the curved VC device are employed in the curved pieces, I think the same behaviour will exist. And the next study will focus on the case of curved pieces. I appreciate the reviewer give our study group a novel topic. The title has revised to “Investigations on Temperatures of the Flat Insert Mold Cavity using VCRHCS with CFD Simulation”, and highlights the “Flat”. I believe that the title now reflects to the real scope of the work.

Reviewer 2 Report
Thank you for submitting your paper. The work done here draws attention to a significant subject temperature evaluation of insert mould cavities. I have found the paper to be interesting. However, several issues need to be addressed properly before the paper is being considered for publication. My comments including major and minor concerns are given below:
Please consider reviewing the abstract and highlight the novelty, major findings, and conclusions. I suggest reorganizing the abstract, highlighting the novelties introduced. The abstract should contain answers to the following questions:
What problem was studied and why is it important?
What methods were used?
What conclusions can be drawn from the results? (Please provide specific results and not generic ones).
The abstract must be improved. It should be expanded. Please use numbers or % terms to clearly shows us the results in your experimental work.
Please consider reporting on studies related to your work from mdpi journals.
Figure 1 is of poor quality, please simply use word file diagrams and move this figure to Materials and methods section.
Line 33 the authors should remove all bulk citations, unless given full credit afterwards. The authors should check for this issue elsewhere in the manuscript.
Figure 2 add scale bar
Figure 5 add (a) and (b)
Figure captions should be improved.
Poorly organised manuscript, figures and layput of different sections.
Authors talk about CFD simulation but where is the model setup and all other details, where is the comparison with experimental data.
Figure 6 move to the results and discussion section.
The title of the manuscript does not reflect that there is simulation work in the manuscript, this must be clearly highlighted in the title.
The manuscript suffers from poor English writing style. Many sentences are difficult to read or interpret.
The authors should add the figure after the first time is being mentioned in a paragraph.
General discussion without any critical evaluation, Not a single reference in the results and discussion section.
The results are merely described and is limited to comparing the experimental observation and describing results. The authors are encouraged to include a more detailed results and discussion section and critically discuss the observations from this investigation with existing literature.
Conclusion can be expanded or perhaps consider using bullet points (1-2 bullet points) from each of the subsections.
Author Response
Response to Reviewer 2 Comments
The author is grateful to the reviewers whose comments have helped me improve the manuscript. Thanks Reviewers for their time and effort for the present manuscript. I appreciate his/her comments that my work presents “Thank you for submitting your paper. The work done here draws attention to a significant subject temperature evaluation of insert mould cavities. I have found the paper to be interesting.”. I have reconstructed and reorganized the whole manuscript carefully. And in order to improve the quality of the paper, the whole manuscript has been checked carefully to avoid any grammar or syntax error. Meanwhile, some statements have been rephrased or refined to improve the language usage in the manuscript. I have accounted and discussed the phenomenon more details in the revised version. I have added more comments to explain the physical effects of the plots. The review comments and descriptions of revisions are listed as following. I have included a separate copy of the revised paper in which I highlighted all the modifications made according to all the reviewers’ comments in RED colour. I believe that the present paper is now acceptable for publication.
Thank you for submitting your paper. The work done here draws attention to a significant subject temperature evaluation of insert mould cavities. I have found the paper to be interesting. However, several issues need to be addressed properly before the paper is being considered for publication. My comments including major and minor concerns are given below:
Point 1: Please consider reviewing the abstract and highlight the novelty, major findings, and conclusions. I suggest reorganizing the abstract, highlighting the novelties introduced. The abstract should contain answers to the following questions:
What problem was studied and why is it important?
What methods were used?
What conclusions can be drawn from the results? (Please provide specific results and not generic ones).
The abstract must be improved. It should be expanded. Please use numbers or % terms to clearly shows us the results in your experimental work.
Response 1: Thanks for the reviewer’s comment. I have been improved the abstract based on the valuable comments of novelties, methods, and specific results. Modifications have been made in the revised manuscript remarked red words.
Point 2: Please consider reporting on studies related to your work from mdpi journals.
Response 2: Thanks for the reviewer’s comment. I have been added some references on studies related to my work from mdpi journals in introductions. They were Refs. [2, 4, 7], [10-13], [16, 17], and [19] as below. Modifications have been made in the revised manuscript remarked red words.
- Surace, R., Basile, V., Bellantone, V., Modica, F., & Fassi, I. (2021). Micro Injection Molding of Thin Cavities Using Stereolithography for Mold Fabrication. Polymers, 13(11), 1848.
- García-Camprubí, M., Alfaro-Isac, C., Hernández-Gascón, B., Valdés, J. R., & Izquierdo, S. (2021). Numerical approach for the assessment of micro-textured walls effects on rubber injection moulding. Polymers, 13(11), 1739.
- Gang, Q., Wang, R. T., & Wang, J. C. (2021). Estimations on Properties of Redox Reactions to Electrical Energy and Storage Device of Thermoelectric Pipe (TEP) Using Polymeric Nanofluids. Polymers, 13(11), 1812.
- Chang, Y. T., Wang, R. T., & Wang, J. C. (2021). PMMA Application in Piezo Actuation Jet for Dissipating Heat of Electronic Devices. Polymers, 13(16), 2596.
- Bula, K., Sterzyński, T., Piasecka, M., & Różański, L. (2020). Deformation mechanism in mechanically coupled polymer–metal hybrid joints. Materials, 13(11), 2512.
- Chen, S. C., Chang, C. W., Tseng, C. Y., Shen, E. N., & Feng, C. T. (2021). Using P (pressure)-T (temperature) path to control the foaming cell sizes in microcellular injection molding process. Polymers, 13(11), 1843.
- Elduque, A., Elduque, D., Pina, C., Clavería, I., & Javierre, C. (2020). Correction: Elduque, A., et al. Electricity Consumption Estimation of the Polymer Material Injection-Molding Manufacturing Process: Empirical Model and Application. Materials, 13(11), 2548.
- Walczak, A., Szypłowska, A., Janik, G., & Pęczkowski, G. (2021). Dynamics of Volumetric Moisture in Sand Caused by Injection Irrigation—Physical Model. Water, 13(11), 1603.
- Alzaabi, M. A., Leon, J. M., Skauge, A., & Masalmeh, S. (2021). Analysis and Simulation of Polymer Injectivity Test in a High Temperature High Salinity Carbonate Reservoir. Polymers, 13(11), 1765.
- Dolza, C., Fages, E., Gonga, E., Gomez-Caturla, J., Balart, R., & Quiles-Carrillo, L. (2021). Development and characterization of environmentally friendly wood plastic composites from biobased polyethylene and short natural fibers processed by injection moulding. Polymers, 13(11), 1692.
Point 3: Figure 1 is of poor quality, please simply use word file diagrams and move this figure to Materials and methods section.
Response 3: Thanks for the reviewer’s comment. I have been utilized simply word file diagrams to redraw the fig. 1 as below and moved it to the Materials and methods section. Modifications have been made in the revised manuscript remarked red words.
Fig. 1 Experimental architecture diagram
Point 4: Line 33 the authors should remove all bulk citations, unless given full credit afterwards. The authors should check for this issue elsewhere in the manuscript.
Response 4: Thanks for the reviewer’s comment. I have been deleted all bulk citations in introductions. Modifications have been made in the revised manuscript remarked red words.
Point 5: Figure 2 add scale bar
Response 5: Thanks for the reviewer’s comment. I have been added scale bar for Figure 2 as below.
|
0 |
|
24 |
|
12 |
Fig. 2 Photo of vapor chamber (V.C.)
Point 6: Figure 5 add (a) and (b)
Response 6: Thanks for the reviewer’s comment. I have been added (a) and (b) of Figure 5 as below.
(a)Without VC (b)With VC
Fig. 5 CFD simulation models
Point 7: Figure captions should be improved.
Response 7: Thanks for the reviewer’s comment. I have been improved the figure captions. Modifications have been made in the revised manuscript remarked red words.
Point 8: Poorly organised manuscript, figures and layput of different sections.
Response 8: Thanks for the reviewer’s comment. I have been improved the manuscript, figures and layout of different sections. Modifications have been made in the revised manuscript remarked red words.
Point 9: Authors talk about CFD simulation but where is the model setup and all other details, where is the comparison with experimental data.
Response 9: Thanks for the reviewer’s comment. I have been added some comparison with experimental data. Modifications have been made in the revised manuscript remarked red words.
Point 10: Figure 6 move to the results and discussion section.
Response 10: Thanks for the reviewer’s comment. I have been moved Figure 6 and relation discussions to the results and discussion section. Modifications have been made in the revised manuscript remarked red words.
Point 11: The title of the manuscript does not reflect that there is simulation work in the manuscript, this must be clearly highlighted in the title.
Response 11: Thanks for the reviewer’s comment. The title has revised to “Investigations on Temperatures of the Flat Insert Mold Cavity using VCRHCS with CFD Simulation”, and highlights the “CFD simulation”. I believe that the title now reflects to the real scope of the simulation work.
Point 12: The manuscript suffers from poor English writing style. Many sentences are difficult to read or interpret.
Response 12: Thanks for the reviewer’s comment. I have reconstructed and reorganized the whole manuscript carefully. And in order to improve the quality of the paper, the whole manuscript has been checked carefully to avoid any grammar or syntax error. Meanwhile, some statements have been rephrased or refined to improve the language usage in the manuscript. I believe that the present paper is now acceptable for publication. Modifications have been made in the revised manuscript remarked red words.
Point 13: The authors should add the figure after the first time is being mentioned in a paragraph.
Response 13: Thanks for the reviewer’s comment. I have been added the figure after the first time is being mentioned in a paragraph. Modifications have been made in the revised manuscript remarked red words.
Point 14: General discussion without any critical evaluation, Not a single reference in the results and discussion section.
Response 14: Thanks for the reviewer’s comment. I have accounted and discussed the phenomenon more details in the revised version. I have added more comments to explain the physical effects of the plots. I believe that the present paper is now acceptable for publication. Modifications have been made in the revised manuscript remarked red words.
Point 15: The results are merely described and is limited to comparing the experimental observation and describing results. The authors are encouraged to include a more detailed results and discussion section and critically discuss the observations from this investigation with existing literature.
Response 15: Thanks for the reviewer’s comment. I have accounted and discussed the phenomenon more details in the revised version. I have added more comments to explain the physical effects of the plots. I believe that the present paper is now acceptable for publication. Modifications have been made in the revised manuscript remarked red words.
Point 16: Conclusion can be expanded or perhaps consider using bullet points (1-2 bullet points) from each of the subsections.
Response 16: Thanks for the reviewer’s comment. I have been revised the conclusion. Modifications have been made in the revised manuscript remarked red words.

Reviewer 3 Report
In general, the article is scientific. It designed and presented a set of local rapid heating mechanism, and explained related phenomena by CFD. Although there are some details should be modified, the main part is logical and substantial. Overall, I would suggest the publication of this manuscript.
Minor comments and suggestions that the authors may want to address.
Comment 1: In the introduction, the welding line is an important problem to be investigated. However, the importance of using CFD simulation is not mentioned, please explain the reason.
Comment 2: For better understanding, the specific size of the parts should be shown in the Methodology and Materials.
Comment 3: In Fig. 5, the label of the components in it should be marked. In addition, in line 254, the description of the figure needs more detailed.
Comment 4: From line 283 to 286, the temperatures of two points of A and B were all 82℃ and 51 ℃ at 60s with VC. One of the temperatures may be without VC, please check it and modify.
Comment 5: In line 422, it is said that the proper venting mechanism can eliminate the V-shaped notch. However, I cannot find the conclusion in the article, please explain the reason of the conclusion.
Comment 6: Recommend the organization of the paper by sections such as 2.1., 3.2., etc. In addition, the meaning of the red words is misunderstanding.

Author Response
Response to Reviewer 3 Comments
The author is grateful to the reviewers whose comments have helped me improve the manuscript. Thanks Reviewers for their time and effort for the present manuscript. I appreciate his/her comments that my work presents “In general, the article is scientific.” and “Overall, I would suggest the publication of this manuscript.”. I have reconstructed and reorganized the whole manuscript carefully. And in order to improve the quality of the paper, the whole manuscript has been checked carefully to avoid any grammar or syntax error. Meanwhile, some statements have been rephrased or refined to improve the language usage in the manuscript. I have accounted and discussed the phenomenon more details in the revised version. I have added more comments to explain the physical effects of the plots. The review comments and descriptions of revisions are listed as following. I have included a separate copy of the revised paper in which I highlighted all the modifications made according to all the reviewers’ comments in RED colour. The English Editing Certification is the “Choice Language Service (群易翻譯顧問有限公司)” in Taiwan as the attached file. I believe that the present paper is now acceptable for publication.
In general, the article is scientific. It designed and presented a set of local rapid heating mechanism, and explained related phenomena by CFD. Although there are some details should be modified, the main part is logical and substantial. Overall, I would suggest the publication of this manuscript.
Minor comments and suggestions that the authors may want to address.
Comment 1: In the introduction, the welding line is an important problem to be investigated. However, the importance of using CFD simulation is not mentioned, please explain the reason.
Response 1: Thanks for the reviewer’s comment. The paper employed the CFD simulation in investigation on VC thermal performance. And some references of numerical methods as [26, 34, 42-44, 46-47] were mentioned welding line and VC effect.
Comment 2: For better understanding, the specific size of the parts should be shown in the Methodology and Materials.
Response 2: Thanks for the reviewer’s comment. Modifications have been made in the revised manuscript remarked red words.
Comment 3: In Fig. 5, the label of the components in it should be marked. In addition, in line 254, the description of the figure needs more detailed.
Response 3: Thanks for the reviewer’s comment. The label of heater and VC had been marked in Fig. 5. Fig. 5 (a) shows a no vapor chamber system meaning that the heat source was direct touched the P20 mold. Fig. 5 (b) is a VC system. Consequently, the VC was placed between the heat source and P20 mold. Modifications have been made in the revised manuscript remarked red words.
- Without VC
(b) With VC
Fig. 5 CFD simulation models
Comment 4: From line 283 to 286, the temperatures of two points of A and B were all 82℃ and 51 ℃ at 60s with VC. One of the temperatures may be without VC, please check it and modify.
Response 4: Thanks for the reviewer’s comment. The simulation results of the two points of A and B were all 82°C at 60 s with VC. The temperatures of the two points of A and B were all 51°C at 60 s without VC. Modifications have been made in the revised manuscript remarked red words.
Comment 5: In line 422, it is said that the proper venting mechanism can eliminate the V-shaped notch. However, I cannot find the conclusion in the article, please explain the reason of the conclusion.
Response 5: Thanks for the reviewer’s comment. The “the proper venting mechanism can eliminate the V-shaped notch” belongs just prediction, this string words have deleted from conclusion. If a proper venting mechanism can be installed in the mold and combined with VCRHCS, the V-shaped notch should be completely eliminated. Modifications have been made in the revised manuscript remarked red words.
Comment 6: Recommend the organization of the paper by sections such as 2.1., 3.2., etc. In addition, the meaning of the red words is misunderstanding.
Response 6: Thanks for the reviewer’s comment. Modifications have been made in the revised manuscript remarked red words.
Round 2
Reviewer 2 Report
Manuscript reads a little better now.
Fig 4 use (a) and (b)
wording in Fig 1 needs moderation
The English of the paper is extremley poor. The paper can not be accepted with such poor language. For example:
Line 207 "The mold temperature was increased to the upon the glass transition temperature" what is meant by upon?
Line 221-222 " The heat source unconnected from the VC at the moment that the filling accomplished" another example of poorly written sentence.
The whole manuscript is not written in a scientific language.
Lines 278-280 do not make sense, what are the authors trying to tell us here?
Lines 388-395 does not read well at all.
A lot of wording does not make sense or is not accurate, for example "the ranges of heat transfer" it is not clear what the authors mean by that. This is just an example and there are many other like this elsewhere in the manuscript.
Author Response
Response to Reviewer 2 Comments
The author is grateful to the reviewers whose comments have helped me improve the manuscript. Thanks Reviewers for their time and effort for the present manuscript. I appreciate his/her comments that my work presents “Manuscript reads a little better now.”. I have reconstructed and reorganized the whole manuscript carefully. And in order to improve the quality of the paper, the whole manuscript has been checked carefully to avoid any grammar or syntax error. Meanwhile, some statements have been rephrased or refined to improve the language usage in the manuscript. I have accounted and discussed the phenomenon more details in the revised version. I have added more comments to explain the physical effects of the plots. The review comments and descriptions of revisions are listed as following. I have included a separate copy of the revised paper in which I highlighted all the modifications made according to all the reviewers’ comments in RED colour. I believe that the present paper is now acceptable for publication.
Manuscript reads a little better now.
Point 1: Fig 4 use (a) and (b)
Response 1: Thanks for the reviewer’s comment. I have been added (a) and (b) of Figure 4 as below. Modifications have been made in the revised manuscript remarked red words.
- Right side (b) Left side
Fig. 4 Actual photograph of this mold of VCRHCS
Point 2: wording in Fig 1 needs moderation
Response 2: Thanks for the reviewer’s comment. Fig. 1 is moderate as below. Modifications have been made in the revised manuscript remarked red words.
Fig. 1 Experimental architecture diagram
Point 3: The English of the paper is extremley poor. The paper can not be accepted with such poor language. For example:
Line 207 "The mold temperature was increased to the upon the glass transition temperature" what is meant by upon?
Line 221-222 " The heat source unconnected from the VC at the moment that the filling accomplished" another example of poorly written sentence.
The whole manuscript is not written in a scientific language.
Lines 278-280 do not make sense, what are the authors trying to tell us here?
Lines 388-395 does not read well at all.
A lot of wording does not make sense or is not accurate, for example "the ranges of heat transfer" it is not clear what the authors mean by that. This is just an example and there are many other like this elsewhere in the manuscript.
Response 3: Thanks for the reviewer’s comment. The English Editing Certification is the “Choice Language Service (群易翻譯顧問有限公司)” in Taiwan as below. Modifications have been made in the revised manuscript remarked red words.

This manuscript is a resubmission of an earlier submission. The following is a list of the peer review reports and author responses from that submission.
Round 1
Reviewer 1 Report
I have previously analyzed this paper and submitted comments for the authors. The authors responded to the comments made and improved the paper accordingly.
Reviewer 2 Report
The manuscript is not a significant contribution to the field of injection molding, because the vapor chamber for rapid heating and cooling system method has been applied for more than 10 years.
The work is not well organized and comprehensively described, because it misses several details on the experimental and numerical analysis, e.g., the governing equations of the process were not given.
The work is not scientifically sound and the English used is not correct and not readable at all.
There is several auto-citations from the authors of works published in International Conferences (Ref 22-25) which are inapropriate since the authors also cite works in peer-reviewed journals.
The work does not fit the high standards to be published in Polymers Journal.
Reviewer 3 Report
Comments[Feedback 1] While it is good to mention the flow of research for Weld Line improvement, it is recommended to provide information on recent studies on VCs or RHCM techniques associated with the subject to enhance understanding of this study.
[Feedback 2]If some give an example of injecting with an actual insert installed, it will show the results of this study more clearly.
[Feedback 3] It is necessary to specify what you want to know through each experiment in Figure 1.
[Feedback 4]The positions of the figure are far from the description in the text.
[Feedback 5] It is recommended to represent the operation sequence of VCRHCS as Figure.
[Feedback 6] In the case of Figure 6, it is necessary to increase the readability of readers by clarifying the display of cycle time and VC status.
[Feedback 7] I compared the temperature change at each location based on time, but the graph is divided, and there is no separate mark at the point to be talked about. It would be nice to have more detailed pointing.
[Feedback 8] Need further clarification on why the temperature rise is higher in the experimental group without VC for 0-30 seconds
[Feedback 9] State the test conditions of each sample in the figure 8 title.
[Feedback 10] (a) to (d) of Figure 9 are not aligned. (c),(d) needs to be explained where it is in the sample and why it is selected.
[Feedback 11] Difficult to determine the exact value due to the error in Table 1.
[Feedback 12] I think it will be helpful to understand if you explain why the hole-shaped specimen was used.
[Feedback 13] Discussion is required for the presented results, and direct evidence (experiment) or academic theory of the cause of the phenomenon is required.
[Feedback 14] If you describe how much the presented results have improved compared to the existing studies, the novelty of the paper can be conveyed well.